# Morphological Evaluation of Bone by CT to Determine Primary Stability—Clinical Study

**DOI:** 10.3390/ma13112605

**Published:** 2020-06-08

**Authors:** Masaaki Takechi, Yasuki Ishioka, Yoshiaki Ninomiya, Shigehiro Ono, Misato Tada, Takayuki Nakagawa, Kazuki Sasaki, Hiroshi Murodumi, Hideo Shigeishi, Kouji Ohta

**Affiliations:** 1Department of Oral and Maxillofacial Surgery, Graduate School of Biomedical and Health Sciences, Hiroshima University, Hiroshima 734-8553, Japan; iss1277@hiroshima-u.ac.jp (Y.I.); yn@hiroshima-u.ac.jp (Y.N.); onoshige@hiroshima-u.ac.jp (S.O.); misatot@hiroshima-u.ac.jp (M.T.); tnakaga@hiroshima-u.ac.jp (T.N.); sasakik@hiroshima-u.ac.jp (K.S.); d165608@hiroshima-u.ac.jp (H.M.); 2Department of Public Oral Health, Graduate School of Biomedical and Health Sciences, Hiroshima University, Hiroshima 734-8553, Japan; shige@hiroshima-u.ac.jp (H.S.); otkouji@hiroshima-u.ac.jp (K.O.)

**Keywords:** dental implants, primary stability, implant stability quotient, computed tomography value

## Abstract

Background: Primary stability is an important prognostic factor for dental implant therapy. In the present study, we evaluate the relationship between implant stability evaluation findings by the use of an implant stability quotient (ISQ), an index for primary stability, and a morphological evaluation of bone by preoperative computed tomography (CT). Subjects and methods: We analyzed 98 patients who underwent implant placement surgery in this retrospective study. For all 247 implants, the correlations of the ISQ value with cortical bone thickness, cortical bone CT value, cancellous bone CT value, insertion torque value, implant diameter, and implant length were examined. Results: 1. Factors affecting ISQ values in all cases: It was revealed that there were significant associations between the cortical bone thickness and cancellous bone CT values with ISQ by multiple regression analysis. 2. It was revealed that there was a significant correlation between cortical bone thickness and cancellous bone CT values with ISQ by multiple regression analysis in the upper jaw. 3. It was indicated that there was a significant association between cortical bone thickness and implant diameter with ISQ by multiple regression analysis in the lower jaw. Conclusion: We concluded that analysis of the correlation of the ISQ value with cortical bone thickness and values obtained in preoperative CT imaging were useful preoperative evaluations for obtaining implant stability.

## 1. Introduction

Diagnostic imaging used to examine the morphology of bone at a planned dental implant site is very useful for implant therapy. Recently, computed tomography (CT) has become indispensable for preoperative diagnosis and is commonly used to evaluate the morphology and bone mass of the jawbone at an implant site [1]. According to the Guidelines for the Use of Diagnostic Imaging in Implant Dentistry 2008 [2], findings obtained with a multiple detector row CT (MDCT) and a cone-beam CT (CBCT), which have become widespread in dental clinics, are considered valid for preoperative diagnosis. This is because they provide accurate information for morphological evaluations and distance measurements of the jawbone [3,4,5]. A correlation between the CT values of the cancellous bone of the jaw and bone quality at the time of implant surgery has been shown [6,7].

For noninvasive evaluation of the intraosseous stability of a dental implant, the implant stability quotient (ISQ) [8] and Periotest [9] values are helpful for evaluating primary stability. A significant correlation was found between a low ISQ value and irretrievably failed implants [10]. Interestingly, preparing the implant bed with an ultrasonic device before tooth root extraction resulted in an increased ISQ value [11]. For a successful implant therapy, the acquisition of primary stability is an important factor that has a decisive influence on long-term prognosis, and continuing the maintenance of reliable osseointegration contributes to successful implant therapy [12,13,14]. Accordingly, evaluation of primary stability by determining the intraosseous stability of the dental implant is necessary for predicting the clinical course.

The initial ISQ value reflects the stiffness of the implant–bone complex [15,16]. Additionally, it is demonstrated that thicker cortical bone leads to a higher ISQ value [17]. Previous studies have examined the impact of bone structure on implant stability in human subjects [16,18,19], and a correlation between the ISQ and the thickness of cortical bone at the time of surgery has been reported. However, there is no report written about the elucidation of the relationship between implant stability evaluation findings by the use of ISQ, an index for primary stability, and morphological evaluation of bone by a preoperative CT in the upper and the lower jaw separately. Elucidation of the relationship between the implant stability evaluation findings by the use of ISQ, an index for primary stability, and morphological evaluation of bone by a preoperative CT are effective for preoperative determination of conditions related to the acquisition of good implant stability. In the present study, we evaluated the condition of bone at planned implant sites obtained by MDCT, using CT results obtained in examinations of cortical and cancellous bone, as well as cortical bone thickness. Furthermore, clinical examinations were performed to determine the correlation between various factors and ISQ values at the time of placement.

## 2. Subjects and Methods

A total of 168 patients, who underwent implant placement surgery at the Department of Oral and Maxillofacial Reconstructive Surgery, Hiroshima University Hospital between September 2013 and March 2017, were enrolled in this retrospective study. The following were considered to be inclusion criteria: patients who were ≥20 years old, partially edentulous with adequate alveolar bone volume for implant insertion, good general health without any uncontrolled systemic diseases, and osteoporosis patients were not included among all patients (168 patients). Then we excluded subjects needing bone grafting due to insufficient bone width and height (n = 43), those with poorly controlled diabetes (n = 2), those with contraindications for minor oral surgical procedures (n = 3), and edentulous people (n = 5). An MDCT scan was performed preoperatively to be required for detailed bone analysis by measuring the CT value (17 patients were excluded after the CBCT scan was performed) and ISQ determination was done on placement. Finally, we analyzed 98 patients in this study (Figure 1).

The present cohort consisted of 36 males (92 implants placed, mean age 59.0 ± 13.6 years) and 62 females (155 implants placed, mean age 56.3 ± 16.8 years), for a total of 98 subjects (247 dental implants placed, mean age 57.3 ± 15.7 years). As for implant sites, 154 were placed in the upper jaw and 93 in the lower jaw (Table 1). All implants were inserted more than 4 months after tooth extraction.

Selected dental implants were NobelReplace Tapered (Nobel Biocare, Gothenburg, Sweden). Table 1 shows the implant length and diameter. All the implants were placed using an Osseo set^TM^ 200 (W & H Dentalwerk Bürmoos GmbH, Austria), according to the manufacture’s recommended protocol. Osstell ISQ^TM^ (Osstell, Integration Diagnostic AB, GoteborgSvagen, Sweden) was used to measure the implant stability in ISQ, by one expert operator. The study design was approved by the Ethical Committee of Hiroshima University (Permission no. E-1528). Following an explanation regarding the content and purpose of the present study, informed consent was obtained from each patient.

### 2.1. CT

An Aquilion ONE^®^ (Systems, Otawara, Tochigi, Japan) and an Aquilion Precision^®^ (Systems, Otawara, Tochigi, Japan) were used for the CT examinations. CT scans were mainly acquired with 160 or 320-detector row CT scanners. The stent for the CT, marked with a gutta-percha point (GP) at the planned implant site, was prepared and appropriately attached to the patient. The stent was fixed firmly by tooth support. Additionally, this stent was converted into an orientation guide for implant surgery.

### 2.2. CT Measuring Method

CT measurements were performed using methods reported by Kumasaka et al. [20] and Fukudome et al. [21]. The thickness of the cortical bone at the implant site was determined based on the thickness of the cortical bone at the alveolar crest of the planned site, marked with a GP. The cortical bone of the alveolar crest at the planned site was measured 3 times, and the mean was calculated (Figure 2). As for the cancellous bone, the mean of the CT values for the 10-mm^2^ region of interest (ROI) in the cancellous bone at the planned site was calculated (Figure 2). In this study, one investigator evaluated the CT values 3 times. The intrarater reliability of the investigator was evaluated using an intraclass correlation coefficient. An investigator calculated the CT data twice, each in 10 different sites. The calculated value of the intraclass correlation coefficient was 0.85, suggesting that the investigator had excellent reliability according to the criteria for intraclass correlation coefficients [22].

### 2.3. CT Value Correction

Differences among devices and imaging conditions can affect the CT values obtained; thus, corrections were performed using the formulae shown following [20]. 

CT measurements at the GP, indicating the planned implant site, were performed 3 times, and the mean was calculated (Figure 2). The CT values for cortical and cancellous bone were converted on the basis of the ratio to the median GP measured in all cases as 3068.83 HU, using the following formulae:Cortical bone value = Cortical bone CT value × 3068.83/GP(1)
Cancellous bone value = Cancellous bone CT value × 3068.83/GP(2)

### 2.4. ISQ

The ISQ measurements were performed using an Osstell ISQ^TM^ in accordance with the method reported by Takechi et al. [23]. A SmartPeg device was attached to the implant body with a wrench at an insertion torque value of 5 Ncm. After setting the tip of the probe vertically to the SmartPeg, the ISQ measurements were performed 3 times at a point 2 mm between the probe and the SmartPeg, and the mean was calculated. The ISQ value is based on the underlying resonance frequency, and ranges from 1 (lowest stability) to 100 (highest stability), with the implant stability increasing as the ISQ value increases. It has been found that the ISQ measurements show a high degree of repeatability (1% variation for individual implants). 

### 2.5. Analysis

For all 247 implants, correlations of the ISQ value with cortical bone thickness, cortical bone CT value, cancellous bone CT value, insertion torque value (ITV), implant diameter, and implant length were examined using Spearman’s rank correlation coefficient. Furthermore, multiple (linear) regression analysis (stepwise selection method) was performed using the ISQ value as the objective variable and factors that showed a significant correlation with the ISQ value as explanatory variables. For all analyses, the statistical level of significance was set at <5%. Based on the difference in anatomical structure morphology between the upper and lower jaws, the present 247 implants were divided into 2 groups, implants of the upper jaw (n = 154) and those of the lower jaw (n = 93). Subsequently, examinations for the association with the ISQ value for both groups were conducted using the same method as used for all of the present dental implants. The sample size required for the correlation using G*Power (version 3.1.9.4, Heinrich-Heine-Universität Düsseldorf, Germany) with a statistical power of 80%, a significance level of 5%, and an effect size of 0.3 was calculated to be 82.

## 3. Results

### 3.1. Factors Affecting ISQ Values in All Cases

Factors that showed a significant correlation with the ISQ value were cortical bone thickness (correlation r = 0.807, *p* < 0.001), cortical bone CT value (r = 0.163, *p* < 0.05), cancellous bone CT value (r = 0.222, *p* < 0.001), ITV (r = 0.355, *p* < 0.001), implant diameter (r = 0.371, *p* < 0.001), and implant length (r = −0.139, *p* < 0.05) (Figure 3). A multiple regression analysis of these six factors was also performed, which revealed significant associations of cortical bone thickness (β = 0.695, *p* < 0.01) and cancellous bone CT (β = 0.132, *p* < 0.05) values with ISQ. The prediction formula was as follows: ISQ value = 35.748 + 32.892 × cortical bone thickness + 0.005 × cortical bone CT value(3)

Thus, it was concluded that cortical bone thickness exerts a large effect on the ISQ value (Table 2).

### 3.2. Factors Affecting ISQ Values in Upper Jaw

Factors in the upper jaw that showed a significant correlation with the ISQ value were cortical bone thickness (r = 0.751, *p* < 0.001), cortical bone CT value (r = 0.170, *p* < 0.05), cancellous bone CT value (r = 0.355, *p* < 0.001), ITV (r = 0.253, *p* < 0.01), and implant diameter (r = 0.200, *p* < 0.05) (Figure 4). Results of the multiple regression analysis of those five factors revealed a significant correlation between cortical bone thickness (β = 0.650, *p* < 0.01) and the cancellous bone CT (β = 0.159, *p* < 0.05) values with the ISQ. The prediction formula was as follows: ISQ value = 30.851 + 36.736 × cortical bone thickness + 0.07 × cancellous bone CT value(4)

These findings also suggested a large effect of cortical bone thickness on the ISQ value (Table 3).

### 3.3. Factors Affecting ISQ Values in Lower Jaw

In the analysis of the lower jaw findings, two factors, cortical bone thickness (r = 0.571, *p* < 0.001) and diameter (r = 0.359, *p* < 0.001), showed a significant correlation with the ISQ (Figure 5). Furthermore, the multiple regression analysis indicated a significant association of cortical bone thickness (β = 0.555, *p* < 0.001) and implant diameter (β = 0.250, *p* < 0.01) with the ISQ. The prediction formula was as follows: ISQ value = 39.808 + 19.868 × cortical bone thickness + 3.138 × diameter(5)

Again, these findings suggested a large effect of cortical bone thickness on the ISQ value (Table 4).

## 4. Discussion

### 4.1. Subjects

The present subjects were patients who underwent an MDCT examination with the use of a stent prior to implant placement and then received surgery. We consider the present cohort to be an unbiased population, as compared to all implant therapy patients treated at our hospital. 

### 4.2. CT Value Correction

In recent years, CT has become indispensable for obtaining preoperative information regarding the jawbone at the planned implant site, for which MDCT and CBCT are often used. CBCT is now widespread, thanks to its advantages, such as low radiation dose and high precision for distance measurements in tomographic images [2]. However, it is unable to obtain CT values that reflect bone density [2,24]. In contrast, MDCT, which has a slightly higher radiation dose as compared to CBCT but shows measurement precision sufficient for preoperative examinations, is considered to be superior for visual evaluations, and can also provide CT values [2]. CT values are used as an index for evaluating bone density [24,25,26]. However, errors can occur depending on the CT device and examination conditions [27]. For this reason, quantitative CT (QCT), using an exclusive phantom, is employed for measuring the bone density on the basis of the CT value obtained [21,28]; however, this is difficult to use clinically as an exclusive phantom is necessary. In the present study, we obtained the CT values with the use of a GP as the diagnostic stent commonly loaded during imaging as a substitute for the default phantom. On the basis of those measurements, the CT values for cortical and cancellous bone were corrected.

### 4.3. Factors Affecting ISQ in Upper Jaw

In the upper jaw, the ISQ values were found to have a correlation with cortical bone thickness, cortical bone CT values, cancellous bone CT values, ITV, and implant diameter. Multiple regression analysis for predicting the ISQ value revealed that the cortical bone thickness and cancellous bone CT value each had a significant effect. CT value is used as an index for speculating jawbone condition, and it has been reported that bone condition can be revealed based on CT values obtained preoperatively for implant therapy [26]. Turkyilmazu et al. [29] examined the relationship between CT values used for expressing the bone condition and mechanical evaluation of an implant body. They showed that ITV and ISQ had a correlation with the CT value of the jawbone at the implant site. Additionally, the present findings suggest that the cancellous bone CT value is a factor useful for predicting ISQ value, though its influence is less than that of cortical bone thickness.

### 4.4. Factors Affecting ISQ Value in Lower Jaw

In the lower jaw, ISQ values showed a correlation with cortical bone thickness and implant diameter; multiple regression analysis also revealed that cortical bone thickness and implant diameter had effects on ISQ. In a study using implant placement models conducted by Arai et al. [30], as the implant diameter increased, the ISQ value also increased. Furthermore, Arai et al. [31] studied the correlation between ISQ value and implant size in patients undergoing implant therapy and reported that the implant length rather than the diameter had an effect on the ISQ value in the upper jaw, while the diameter had an effect on the ISQ in the lower jaw. In the present study, no significant correlation was observed between ISQ values and implant length in either the upper or lower jaw. However, in the lower jaw, the diameter was suggested to be involved as a factor for predicting the ISQ. Thus, the factors with a significant effect on the ISQ value were different between the upper and lower jaws. We considered that these results were caused by differences in the anatomical structure between the jaws, including a lower level of cancellous bone density^14^ and thinner cortical bone [32] in the upper jaw.

Miyamoto et al. [19] determined the cortical bone thickness on the basis of CT images and the relationship with the ISQ value. They reported that cortical bone thickness and ISQ had a strong positive correlation. The results of the present study also showed a positive correlation between ISQ values and cortical bone thickness in both the upper and lower jaws, suggesting that this factor has a great effect on predicting ISQ. As higher ISQ values (i.e., >60 ISQ) are associated with increased implant stability [33], we predict that we can get enough primary stability if the cortical bone thickness is more than 1 mm from the results of the correlation analysis between ISQ value and cortical bone thickness. Therefore, it is very important to get enough primary stability so that we can evaluate cortical bone thickness before surgery. 

In the present study, we also examined primary stability. Some cases showed a high ISQ value clinically at the time of the implant placement but had a poor prognosis because of an implant falling out. In contrast, others with not so high ISQ values at implant placement ultimately showed a favorable prognosis with high implant stability. 

There are some limitations of the study. It is thought that the primary stability depends not only on the recipient site but also on the surgical technique and implant design, as well as various other factors [33,34,35,36,37]. In the present study, all the implants used were Nobel Replace Tapered (bone level type implants). Therefore, the results of this study were only for tapered, internal connection implant design. Additionally, although one surgeon inserted the implants in one facility in this study, it might be necessary to collect more data across multi-facilities to examine the generalizability of the results.

The ISQ analysis can supply clinically relevant information about the condition of the implant at any stage of the treatment or at follow-up examinations. Many reports show that implants with a high ISQ value during follow-up examinations are successfully integrated, while low and decreasing ISQ values may be a sign of ongoing implant failure [32,33]. Although all implants have not lost so far in this study, it will be necessary to follow patient courses for a longer period by measuring the ISQ value in order to examine changes over time more fully.

## 5. Conclusions

In this paper, the relationship between the implant stability evaluation findings by the use of ISQ, an index for primary stability, and morphological evaluation of bone by a preoperative CT was evaluated. The main conclusions can be drawn as follows: (1)Factors that showed a significant correlation with the ISQ value in all subjects were cortical bone thickness, cortical bone CT value, cancellous bone CT value, ITV, and implant diameter and length. Multiple regression analysis, using the ISQ value as the objective variable, revealed that cortical bone thickness and cancellous bone CT value had a significant association with ISQ. These results indicated that cortical bone thickness has a great effect on predicting the ISQ value.(2)In the upper jaw, cortical bone thickness, cortical bone CT value, cancellous bone CT value, ITV, and implant diameter showed a significant correlation with ISQ. Multiple regression analysis, using the ISQ value as the objective variable, revealed a significant association of cortical bone thickness and cancellous bone CT value with ISQ. These results also indicated a significant effect of cortical bone thickness for predicting the ISQ value.(3)In the lower jaw, cortical bone thickness and implant diameter were confirmed to be significantly correlated with ISQ. Multiple regression analysis, using the ISQ value as the objective variable, also showed that both had a significant association with ISQ. Again, the cortical bone thickness was indicated to have a significant effect as a factor for predicting the ISQ value.

## Figures and Tables

**Figure 1 materials-13-02605-f001:**
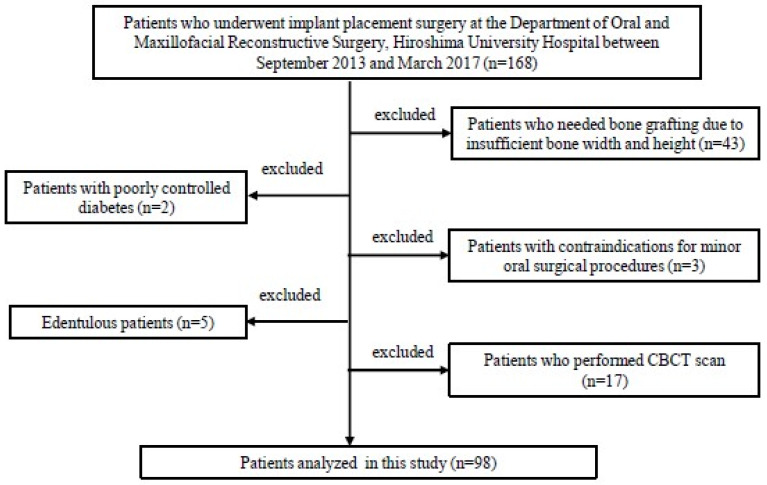
Selection of patients.

**Figure 2 materials-13-02605-f002:**
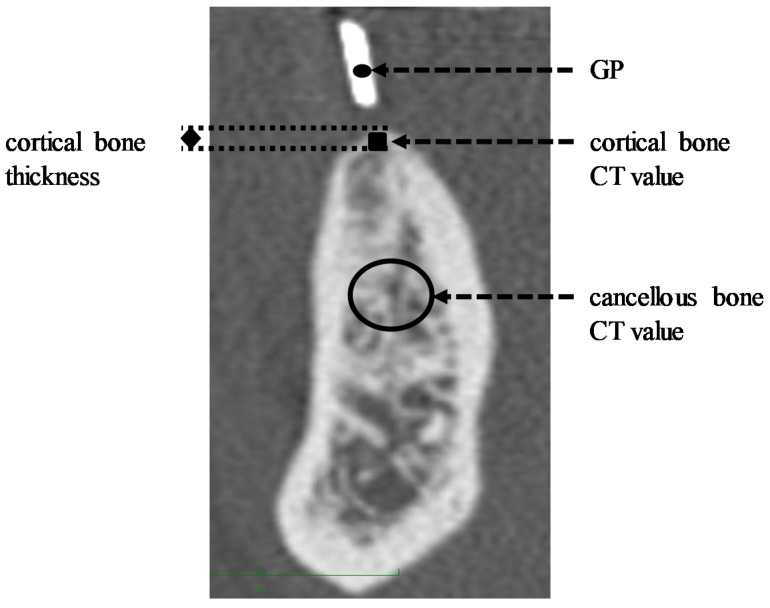
Computed tomography (CT) measurement methods. Gutta-percha point (GP): CT value at gutta-percha point. Cortical bone thickness: the thickness of cortical bone at alveolar crest. Cortical bone CT value: mean of 3 CT measurements of the cortical bone at alveolar crest. Cancellous bone CT value: mean CT value for 10 mm^2^ ROI set in the cancellous bone.

**Figure 3 materials-13-02605-f003:**
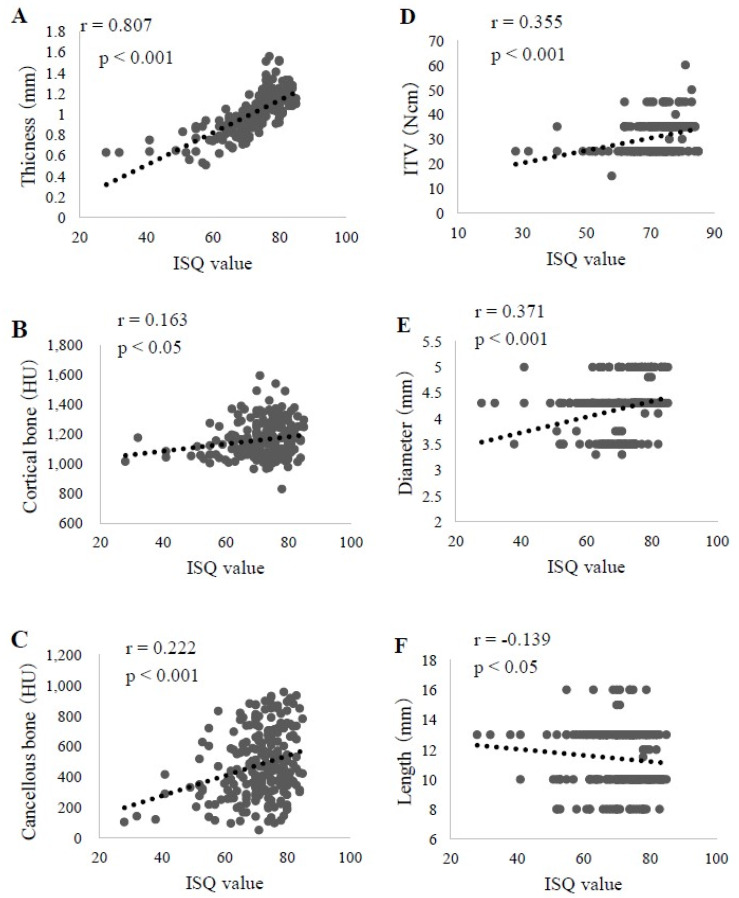
Correlation of each examined factor with the implant stability quotient (ISQ) value in all subjects (n = 247). **A**. Cortical bone thickness. **B**. Cortical bone CT value. **C**. Cancellous bone CT value. **D**. Insertion torque value (ITV). **E**. Implant diameter. **F**. Implant length.

**Figure 4 materials-13-02605-f004:**
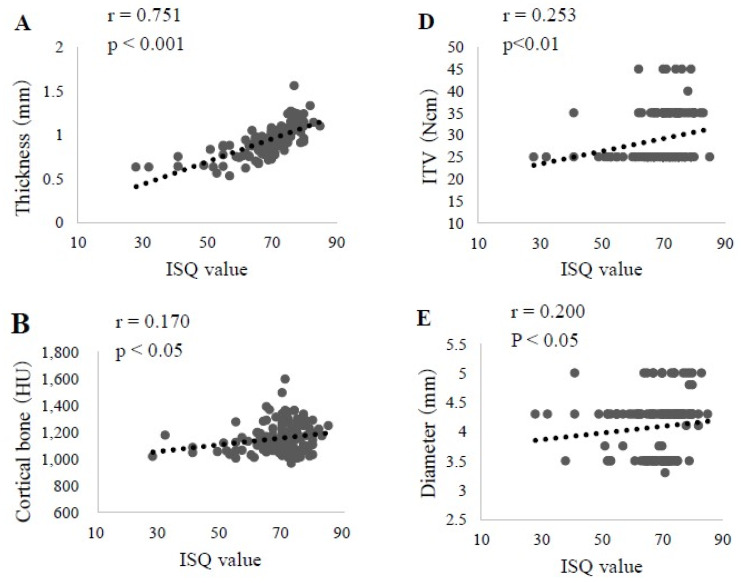
Correlation of each examined factor with the ISQ value in upper jaws (n = 154). **A**. Cortical bone thickness. **B**. Cortical bone CT value. **C**. Cancellous bone CT value. **D**. ITV. **E**. Implant diameter. **F**. Implant length.

**Figure 5 materials-13-02605-f005:**
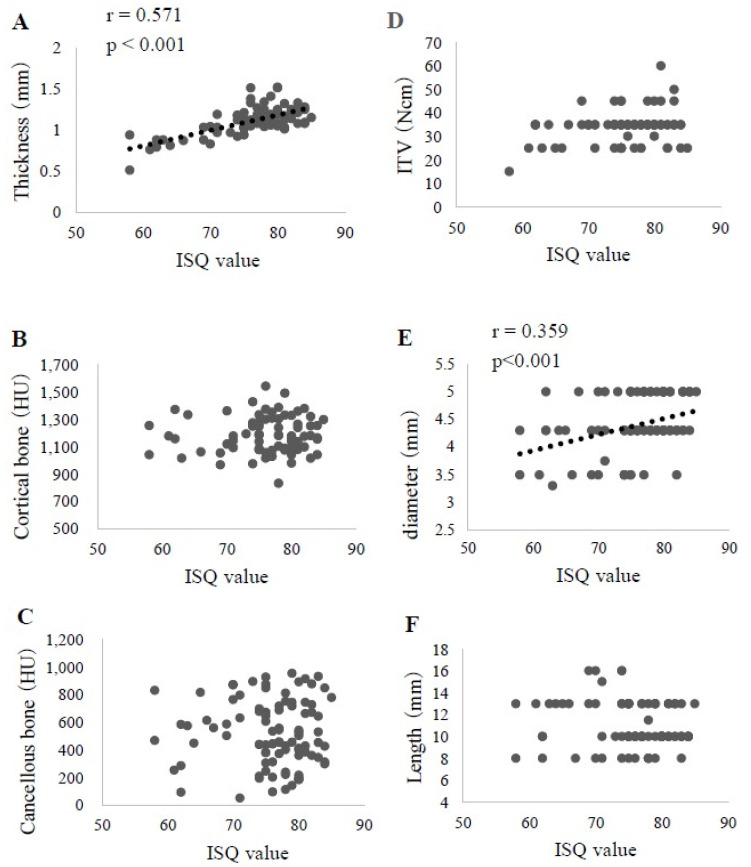
Correlation of each examined factor with the ISQ value in lower jaws (n = 93). **A**. Cortical bone thickness. **B**. Cortical bone CT value. **C**. Cancellous bone CT value. **D**. ITV. **E**. Implant diameter. **F**. Implant length.

**Table 1 materials-13-02605-t001:** Subjects.

		**Male**	**Female**	**Total**
Cases of Implant Treatment		36	62	98
Number of Implants		92	155	247
		**Maxilla**	**Mandible**	**Total**
Number of Implants		154	93	247
**Implant Length**	**Implant Diameter**			**Total**
	**Narrow (3.5 mm)**	**Regular (4.3 mm)**	**Wide (5.0 mm)**	
8 mm	5	17	4	26
10 mm	22	48	28	98
13 mm	32	65	15	112
16 mm	7	4	0	11
Total	66	134	47	247

**Table 2 materials-13-02605-t002:** Multiple regression analysis of all subjects using the ISQ value as an objective variable.

Explanatory Variable	Unstandardized Coefficient		Standardization	*p* Value	95% Cofidence Interval
	B	Standard Error	β		
Constant	35.748	2.544			
Cortical bone thickness (mm)	32.892	2.404	0.695	0.000	28.151–37.634
Cancellous bone CT value (HU)	0.005	0.002	0.132	0.010	0.001–0.009

(n = 247).

**Table 3 materials-13-02605-t003:** Multiple regression analysis of the upper jaw using the ISQ value as an objective variable.

Explanatory Variable	Unstandardized Coefficient		Standardization	*p* Value	95% Cofidence Interval
	B	Standard Error	β		
Constant	30.851	3.694			
Cortical bone thickness (mm)	36.736	3.848	0.650	<0.001	29.114–44.358
Cancellous bone CT value (HU)	0.007	0.003	0.159	0.021	0.001–0.013

(n = 154).

**Table 4 materials-13-02605-t004:** Multiple regression analysis of the lower jaw using the ISQ value as an objective variable.

Explanatory Variable	Unstandardized Coefficient		Standardization	*p* Value	95% Cofidence Interval
	B	Standard Error	β		
Constant	39.808	4.928			
Cortical bone thickness (mm)	19.868	3.300	0.555	<0.001	13.293–26.442
Cancellous bone CT value (HU)	3.138	1.157	0.250	0.008	0.834–5.442

(n = 93).

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
