# Peer review of "Morphological Evaluation of Bone by CT to Determine Primary Stability—Clinical Study"

_materials, 2020, doi:10.3390/ma13112605_

Round 1

Reviewer 1 Report

The authors of the manuscript compare the primary stability of dental implants with the bone density measured in a preoperative CT. The topic itself is not new. Various studies on this topic are presented in the literature. Therefore, the authors should emphasize the innovations in their study.

In addition, they should always keep in mind that primary stability depends not only on the bone quality of the recipient site but also on the surgical technique and implant design, as well as various other factors.

Abstract

The abstract is too long and should be shortened, especially the introduction and materials and methods part of the abstract. E.g. points such as the inclusion period, informed consent should only be mentioned later in the materials and methods section of the main text.

Introduction:

Line 60: The part of the sentence “MDCT can confirm CT results” creates more confusion than it helps in this context and thus it should be removed.

Line 64: I recommend removing “extremely”

Materials and methods:

What was the study design? Prospective? Was a sample size calculation performed?

What were the inclusion and exclusion criteria of the patients (bone dimension? Indication (single tooth / edentulous, etc)? Were patients with need of bone augmentation procedures excluded? Were patients excluded that required osteotome site preparation? When were implants inserted after tooth extraction (4 weeks? 8 weeks? Healed ridges?)

What type of implants were used? What drilling protocol was used?

The authors should always keep in mind that primary stability also depends on the surgical technique and implant design and not only on the recipient site. This should also be mentioned in the discussion section of the manuscript.

Please correct 1 table: It should be “male” and “female”

Line 91: How was the stent for the CT retained? Tooth support or mucosa support? Was this stent later converted to an orientation guide for implant surgery?

Was the implant insertion torque recorded as well? If yes, this data should be included.

Who performed the measurements? How many examiners performed the measurements? Were they involved in the treatment of patients? Were they calibrated?

How long was the follow-up? Were implants lost?

Discussion:

The part of the sentence “at the Department of Oral and Maxillofacial Reconstructive Surgery, Hiroshima University Hospital” (lines 188- 189) should be deleted as this was already mentioned in the M&M section. The following sentences (lines 189-194) do not provide helpful additional information for this study and should, therefore, be deleted.

The limitations should be discussed in more detail. Please note the points mentioned in the M&M section.

Conclusion:

Lines 246-250 do not contain any new information and should be deleted. This information is already included in other parts of the manuscript

The information given should be presented in a more condensed form. It should not be a complete repetition of the results.

Author Response

Thank you so much for reviewing my paper.

I replied to your comments and revised my manuscript.

Please see and check the attached file.

Reviewer 2 Report

I appreciate the significant amount of time and effort you have placed into making the academic plan and writing your article. The content is interesting, but has insufficient clinical and/or scientific impact to be suitable for materials. The manuscript does not contain IRB information on Research Involving Human Subjects.

  1. The abstract and conclusion should be simple and clear, but it is too long and boring.
  2. In the introduction, there should be mention of the risk of radiation and efforts to minimize it, and at the end, it is good to mention the hypothesis of this study.
  3. Describe the information of ISQ instrument in line of 79.
  4. In line of 91, I think that fixation of the stent would influence the analysis of the results. Explain in detail how you fixed it.

Author Response

(The authors gave the same response as above.)

Reviewer 3 Report

the manuscript is well written and is very interesting for
readers of the journal. It would be advisable to improve the introduction since it is very short and is not proportional to the rest of the sections. It is necessary to specify if the investigation has passed through the ethics committee of the institution or if it has been registered in the clinical trial. the rest of the sections are very well written with very conclusive data. It would be interesting, based on the obtained results, to write a clinical implication in the final part of the discussion section.

Author Response

(The authors gave the same response as above.)

Round 2

Reviewer 1 Report

The revision process improved the manuscript, but there are still factors requiring further clarification.

Furthermore, I am missing a detailed response to the reviewers. In the according link I only receive a pdf with the figures.

I still miss the information on the study design (prospective / retrospective?)

Please describe the inclusion and exclusion criteria for this study in the M&M section.

Depending on the design please use a CONSORT or a STROBE checklist to avoid any further missing information. Additionally use the proposed structure and order. Add a filled out form for the revision.

https://www.strobe-statement.org/index.php?id=available-checklists

http://www.consort-statement.org

The authors should still work on the point limitations in the discussion (line 288-289).

Line 293 should be deleted.

Author Response

Responses to comments raised by Reviewer #1

We are most grateful for the insightful comments and suggestions from the reviewer #1: they helped us greatly improve our paper. As indicated in the responses below, we have taken all those comments into account in the revised version of the manuscript. Changed sections in the revised manuscript appear in red.

Reviewer 1
1. The revision process improved the manuscript, but there are still factors requiring further clarification. Furthermore, I am missing a detailed response to the reviewers. In the according link I only receive a pdf with the figures.

Response

I apologize for missing a detailed response to the reviewers. I attached responses to comments raised by reviewer as described below

2. I still miss the information on the study design (prospective / retrospective?)

Response

We conducted retrospective study.

I described about retrospective study in the abstract and M&M section.

3. Please describe the inclusion and exclusion criteria for this study in the M&M section.

Response

As you indicated, I showed “selection of patients” in figure 1 and described about that in the M&M section.

4. Depending on the design please use a CONSORT or a STROBE checklist to avoid any further missing information. Additionally use the proposed structure and order. Add a filled out form for the revision.

Response

We are grateful for your comments. In accordance with your suggestion, we have followed the comments of reviewer and checked STROBE checklist.

5.The authors should still work on the point limitations in the discussion (line 288-289).

 Response

I described the point limitations in the discussion section.

6.Line 293 should be deleted.

Response

I deleted the sentence of Line 293 as you mentioned.

Reviewer 2 Report

Thank you for correction. I was quietly surprised with the correction method used because you didn't even write a basic revision letter. Reviewers need a revision letter (line by line) to see how all reviewers’ comments have been modified by the author.

As other reviewer pointed out, there have been many papers on the relationship between bone and stability. There is nothing new in the paper now. Although this study was based on data from many patients for 5 years, the strengths of this study alone are not described. For example, if you put results that compare between this results and the data after 5 years of implant, you may have the strength of this study.

Here are my comments in further details:

  1. Previous papers on implant stability and bone were not introduced in the introduction.
  2. Explain why you chose MDCT with a high dose of radiation and what kind of MDCT (the number of channels) you used.
  3. It should be revealed that the results of this study were only for tapered, internal connection implant design.
  4. Make a table on the length and diameter of the implant used.
  5. Present clinical implications and give meaning.
  6. Who evaluated the MDCT ’s data? How many investigators? Was there inter and intra reliability checked?

Author Response

Responses to comments raised by Reviewer #2

We are most grateful for the insightful comments and suggestions from the reviewer #2: they helped us greatly improve our paper. As indicated in the responses below, we have taken all those comments into account in the revised version of the manuscript. Changed sections in the revised manuscript appear in red.

Reviewer 2

As other reviewer pointed out, there have been many papers on the relationship between bone and stability. There is nothing new in the paper now. Although this study was based on data from many patients for 5 years, the strengths of this study alone are not described. For example, if you put results that compare between this results and the data after 5 years of implant, you may have the strength of this study.

Response

I think that what you pointed out is very important. Unfortunately, this time is retrospective study. Therefore, we are going to evaluate the comparison between this result and the data after 5 years of implant.

  1. Previous papers on implant stability and bone were not introduced in the introduction.

Response

I mentioned about implant stability and bone of previous papers in the introduction section.

  1. Explain why you chose MDCT with a high dose of radiation and what kind of MDCT (the number of channels) you used.

Response

MDCT has a high dose of radiation as you mentioned, however we wanted to measure accurate CT value using MDCT and evaluated the relationship between bone quality and primary implant stability. In short, we cannot measure CT value using CBCT correctly.I described “We used an Aquilion ONE® (Systems, Otawara, Tochigi, Japan) and Aquilion Precision® (Systems, Otawara, Tochigi, Japan) for the CT examinations. CT scans were mainly acquired with 160 or 320-detector row CT scanners.” in the M&M section.

  1. It should be revealed that the results of this study were only for tapered, internal connection implant design.

Response

As you indicated, I described about implant design in the M&M and discussion section.

  1. Make a table on the length and diameter of the implant used.

Response

I added table 1 on the length and diameter of the implant used in the M&M section.

  1. Present clinical implications and give meaning.

Response

As you mentioned, I added clinical implications in the discussion section.

    6. Who evaluated the MDCT ’s data? How many investigators? Was there inter and intra reliability checked?

Response

I described as follows in the M&M section.

In this study, one investigator evaluated CT values 3 times.

The intra-rater reliability of the investigator was evaluated using an intraclass correlation coefficient. An investigator calculated the CTs data twice each in 10 different sites. The calculated value of the intraclass correlation coefficient was 0.80, suggesting that the investigator had excellent reliability according to the criteria for intraclass correlation coefficients (Coppieters, Stappaerts, Janssens & Jull, 2002).

Coppieters, M., Stappaerts, K., Janssens, K., & Jull, G. (2002). Reliability of detecting 'onset of pain' and 'submaximal pain' during neural provocation testing of the upper quadrant. Physiother Res Int, 7, 146-156.

Round 3

Reviewer 1 Report

The revision process has further improved the manuscript. Most points have been clarified, but I still miss the inclusion criteria. Exclusion criteria were added as requested.

Author Response

Responses to comments raised by Reviewer #1

We are most grateful for the insightful comments and suggestions from the reviewer #1: they helped us greatly improve our paper. As indicated in the responses below, we have taken all those comments into account in the revised version of the manuscript. Changed sections in the revised manuscript appear in red.

Reviewer 1
1. The revision process has further improved the manuscript. Most points have been clarified, but I still miss the inclusion criteria. Exclusion criteria were added as requested.

Response

I described the inclusion criteria in the M&M section.

Reviewer 2 Report

I am grateful that the research paper has been revised well with much effort.

However, the abstract section does not briefly describe. According to materials regulations, the summary should be shortened to within 200 words.

Please add on subject and method if osteoporosis patients are included in this study

Lastly, I hope that the conclusion section will be explained more briefly for the reader.

Author Response

Responses to comments raised by Reviewer #2

We are most grateful for the insightful comments and suggestions from the reviewer #2: they helped us greatly improve our paper. As indicated in the responses below, we have taken all those comments into account in the revised version of the manuscript. Changed sections in the revised manuscript appear in red.

Reviewer 2

  1. However, the abstract section does not briefly describe. According to materials regulations, the summary should be shortened to within 200 words.

Response

As you indicated, I shortened the sentences of abstract section.

  1. Please add on subject and method if osteoporosis patients are included in this study.

Response

I added that “osteoporosis patients were not included among all patients (168 patients)” in the subject and method.

  1. Lastly, I hope that the conclusion section will be explained more briefly for the reader.

Response

As you mentioned, I revised the conclusion section briefly for the reader.
